# A comprehensive investigation of the reaction behaviorial features of coke with different CRIs in the simulated cohesive zone of a blast furnace

Qing Q. Lv[1,2], Yong S. Tian[1,2], Jun L. Zhou[3], Hua W. Ren[4], Guang H. Wang[1,2]*

1 School of Chemistry and Chemical Engineering, Wuhan University of Science and Technology, Wuhan, Hubei, China, 2 Key Laboratory of Hubei Province for Coal Conversion and New Carbon Materials, Wuhan, Hubei, China, 3 Iron and Steel Research Institute of Shasteel, Zhangjiagang, Jiangsu, China, 4 Coking Division, Xingtai Risun Science and Technology Co., Ltd., Xingtai, Hebei, China

* wghwang@263.net

**Data Availability Statement:** All relevant data are within the paper and its Supporting Information files.

## Abstract

The reaction characteristics and mechanism of coke with different coke reactivity indices (CRIs) in the high-temperature zone of a blast furnace should be fully understood to correctly evaluate the coke quality and optimize ironmaking. In this work, low-CRI coke (coke A) and high-CRI coke (coke B) were charged into a thermogravimetric analyzer to separately study their microstructural changes, gasification characteristics, and reaction mechanism under simulated cohesive zone conditions in a blast furnace. The results show that both coke A and coke B underwent pyrolysis, polycondensation, and graphitization during the heat treatment. The pyrolysis, polycondensation, gasification speed, and dissolution speed rates of coke B were higher than those of coke A. Direct and indirect reduction between sinter and coke occurred in the cohesive zone and had different stages. The consumption rate of coke B was faster than that of coke A during the coke–sinter reduction. The carbon molecules of coke A must absorb more energy to break away from the skeleton than those of coke B.

## 1 Introduction

Metallurgical coke mainly functions as a reducing agent, structural support medium, fuel, and carburant in a blast furnace [1,2]. With the development of ironmaking technology, blast furnaces tend to be used on a large scale, and the pulverized coal injection rate gradually increases. These factors cause coke to withstand intense thermal, mechanical, and chemical stresses in blast furnaces. Coke also supports the entire media in blast furnaces and participates in complicated multiple-phase reactions, especially in cohesive zones, because sinter, pellet, and ore begin to melt at high temperatures. In addition, coke acts as the only solid material that greatly affects the gas distribution and stability of blast furnaces [3,4]. Consequently, the role of coke as a structural support medium enhances. The coke reactivity index (CRI) reflects the ability to resist $CO_2$ dissolution, while the coke strength after a reaction (CSR) indicates

**Funding:** This work was financially Supported by National Key R&D Program of China (2017YFB0304300&2017YFB0304303). The funder is National Key R&D Program of China. And no sponsors of funder play any role in the study design, data dollection and analysis, desision to publish, or preparation of the manuscript. The National Key R&D program provided support in the form of salaries for authors [QQL, YST, JLZ, HWR and GHW]. The Key Laboratory of Hubei Province for Coal Conversion and New Carbon Materials provided support for this study in the form of salary for authors YST and GHW. Iron and Steel Research Institute of Shasteel provided support in the form of salary for author JLZ. Zhangjiagang and Xingtai Risun Science and Technology Co., Ltd. provided support in the form of salary for author HWR. The funders had no role in study design, data collection and analysis, decision to publish, or preparation of the manuscript. The specific roles of these authors are articulated in the 'author contributions' section.

the ability to resist powdering after $CO_2$ dissolution. The CRI shows a good negative correlation with the CSR. Therefore, the resistance of low-CRI coke to fine generation and accumulation is higher than that of high-CRI coke in blast furnaces. It is believed that the low-CRI coke can better support the burden in blast furnaces than high-CRI coke [5,6]. Blast furnace production declines and the fuel ratio increases when the coke CRI increases [7,8]. Previous studies show that when the CRI increases by 1%, the fuel ratio increases by 1.1%, and production decreases by 1.2% [9]. Kurunov [10] considered that the effect of coke quality on blast furnaces was related to production intensity, and this effect enhanced when the production intensity was high. The CRI test method only reflects the reaction of $CO_2$ with coke at 1100˚C, but the actual environment in a blast furnace is more complex than the test condition. CRI test conditions considerably differ from actual conditions in blast furnaces [11–13]. Many extensive studies have been performed to explore the CRI and metallurgical coke degradation mechanism in blast furnaces. Hu [14] and Guo [15] showed that CRI was no longer suitable for blast furnace operation. Sunahara [16] simulated the drip zone reaction of coke in blast furnaces and revealed that coke consumption was slightly related to the CRI. Wang [17] proposed a new testing and evaluating method of cokes with greatly varied CRI and CSR. Research shows that the coke reaction rate at the coke weight loss of 25% (CRR25) and coke strength after reaction (CSR25) can explain the stable operation of a blast furnace. Lyalyuk [18] indicated that the CRI should be determined under specific operating conditions in blast furnaces. Similar results suggest that an extremely low CRI can reduce the thermal efficiency of blast furnaces, and this condition is not conducive to iron oxide reduction [19,20]. Evaluating the quality of coke using the CRI remains controversial because the deterioration of coke in the high-temperature zone of blast furnaces cannot be observed due to the harsh environment and limited access to the low zone of blast furnaces. The deterioration mechanism of coke has not been well explained [21–24].

Coke is simultaneously affected by high-temperature heat, gasification, and ore oxidation in the blast furnace. The current studies have not distinguished how these effects affect the coke structure. There is no distinction between structural changes of coke with different CRIs after they have been subjected to these effects. To study the mechanism by which CRI affects the deterioration of coke in the middle and lower parts of the blast furnace, it is necessary to separately explain the mechanism of heat, gasification, and ore oxidation that affect the structure and consumption of coke of different CRIs. In the present study, thermogravimetric analysis was conducted to simulate the conditions of the cohesive zone in a blast furnace. Low-CRI coke (coke A) and high-CRI coke (coke B) were selected to examine the microstructural changes, gasification characteristics, and reaction mechanism under the simulated cohesive zone conditions in the blast furnace. Chemical composition analysis, scanning electron microscopy, energy spectrum analysis, mass spectrometry, and endothermic differences during the reaction were also performed to describe the chemical, physical, and structural characteristics of coke.

## 2 Experimental

### 2.1 Materials

**(1) Coke.** Coke A with 20.5% CRI and coke B with 28.6% CRI were selected for this experiment. The coke properties are shown in Table 1. Coke was crushed to facilitate an STA test. Then, coke with a particle size of 0.2–0.5 mm was screened, baked at 105±5˚C for more than 2 h to remove moisture, sealed in plastic bags, and stored for later use.

**(2) Sinter.** The chemical composition of the sinter is shown in Table 2. The sinter was crushed to less than 0.1 mm to facilitate the STA test. Then, it was baked at 105±5˚C for more than 2 h to remove moisture, sealed in plastic bags, and stored for later use.

**Table 1. Properties of coke A and coke B.**

| Sample | $M_{40}$/% | $M_{10}$/% | CRI/% | CSR/% | $V_{ad}$/% | $S_{t,d}$/% | $A_d$/% |
|--------|--------|--------|-------|-------|--------|---------|--------|
| Coke A | 89.5 | 5.7 | 20.5 | 69.5 | 2.0 | 0.7 | 12.0 |
| Coke B | 85.2 | 7.2 | 28.6 | 65.5 | 2.5 | 0.7 | 12.0 |

## 2.2 Test methods

TGA was conducted to separately simulate the cohesive zone conditions including high-temperature heat, gasification, and sinter oxidation. The microstructural changes in coke A and coke B after heat treatment were analyzed, and gasification characteristics were detected. The reaction behavior of coke with sinter was investigated, and the consumption rate and structural differences between coke A and coke B were compared and examined.

**(1) Heat treatment of coke under the temperature condition in the cohesive zone.** In this procedure, 25 mg of coke sample was charged into a thermogravimetric analyzer (Netzsch STA 449f) for testing. The coke sample was initially heated from ambient temperature to 750°C at a heating rate of 30°C/min and an $N_2$ flow rate of 30 mL/min. Subsequently, the sample was further heated to 1350°C at a heating rate of 10°C/min and an $N_2$ flow rate of 30 mL/min. After the temperature reached 1350°C, which is similar to the temperature range of the cohesive zone in the blast furnace, it was held constant for 15 min. During the heating process, the coke mass loss was determined based on the abrupt changes in the TGA curves.

After the heat treatment test, the coke sample was removed and crushed to <0.074 mm for a crystal structure change test using an XRD (Shimadzu XRD-6100).

**(2) Reaction test of coke with sinter.** The ratio of coke to iron-bearing minerals (sinter, ore, and pellet) in a blast furnace is usually approximately 1:3. In our experiment, 10 g of coke and 30 g of sinter were uniformly mixed. Subsequently, 25 mg of the sample was obtained and charged into the thermogravimetric analyzer. A reduction reaction between coke and sinter occurred during heating in the thermogravimetric analyzer. The heating procedures were identical to those in the heat treatment test.

CO and $CO_2$ formed during the reaction between coke and sinter, which caused a mass loss of the reactant. A mass loss curve (TGA) was used to study the reaction behavior between sinter and coke. The reaction rate curve (DTG) was the derivative of the TG of the mass loss curve, which indicates the reaction speed. The differential thermal curve (DTA) records the heat absorption and release process. The CO and $CO_2$ compositions were analyzed with a mass spectrometer (QMS403 Aeolos Quadro), which was connected to the thermogravimetric analyzer. The amount of molecule is related to the electron signal intensity; i.e., a stronger signal intensity corresponds to a larger amount of formed gas.

**(3) Analysis of coke morphology.** After the sinter and coke reaction test, the reaction product was obtained for the morphological analysis. First, the surface microstructure was observed in detail by using a scanning electron microscope. Then, the reaction product was then removed from the scanning electron microscope and gently polished gently on a polishing machine. After a plane was ground, the internal morphology and chemical composition of the reaction product were analyzed with a scanning electron microscope (EDS, JIB-4700F).

**Table 2. Chemical composition of the sinter.**

| Sample | Total Fe/% | FeO/% | $SiO_2$/% | $Al_2O_3$/% | CaO/% | MgO/% | P/% | S/% |
|--------|-----------|-------|---------|-----------|-------|-------|-----|-----|
| Sinter | 57.62 | 9.62 | 4.55 | 1.85 | 10.28 | 1.71 | 0.059 | 0.018 |

## 3 Results and discussion

### 3.1 Effect of the high-temperature heat treatment on the graphitized structure of coke

**(1) Coke pyrolysis and polycondensation property analysis.** The mass loss curves of coke A and B are shown in Fig 1. The TGA curves of the two types of coke tend to be uniform. The mass loss increases when temperature increases. A higher temperature corresponds to faster mass loss. The residual masses of coke A and coke B at the end of the reaction are 92.5% and 88.6%, respectively. The result reveals that the mass loss by coke B is greater than that by coke A after the heat treatment. According to coking theory [25], pyrolysis and polycondensation occur during coking. In the low temperature range of <600°C, side chain molecules fall off from aromatic ring compounds, and coke loses mass, as shown in Eq (1). Polycondensation mainly occurs at the high temperature range of >600°C. This change is characterized by dehydrogenation, aromatic structure condensation, and increased aromatic layers, as shown in Eq (2). Thus, the composition of coke continues to change during heating in a blast furnace.

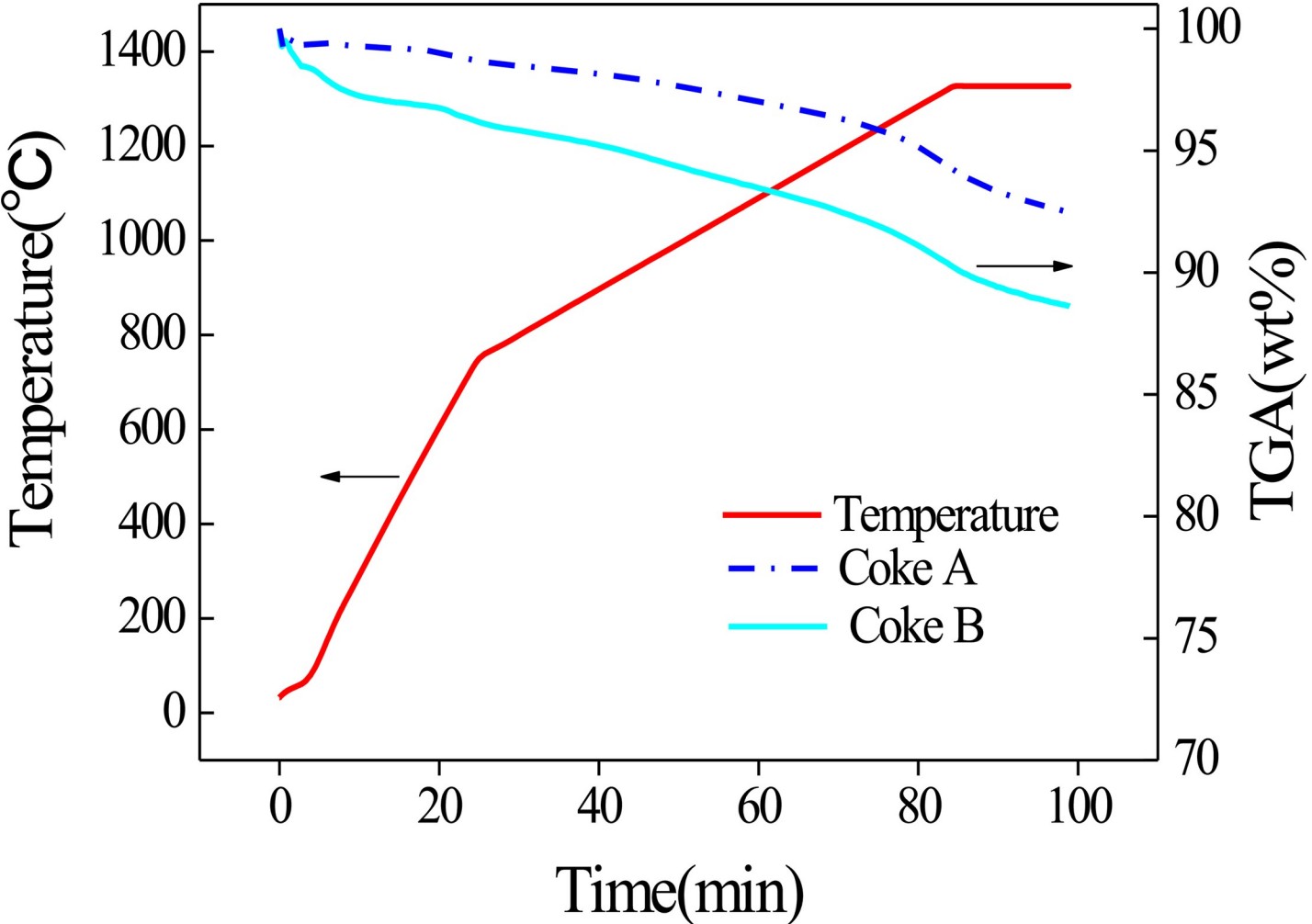

**Fig 1. TGA curve of coke.**

Organic molecules constantly remove small molecules, whereas aromatic structures constantly shrink.

The amount of mass loss of coke B is more than that of coke A, which indicates that the amount of mass lost by coke B is more than that of coke A when they are reheated in the blast furnace. The thermal stability of organic compounds depends on the chemical bond energy of each atom in a molecule. The chemical bond with more energy cannot be easily broken. Among the bond energies of organic compounds, the bond energy of aromatic carbon is the highest, whereas the bond energy of aliphatic carbon is the lowest [26]. The proportion of aliphatic carbon bonds in low-rank coal is higher than that in high-rank coal. Therefore, the thermal stability of low-rank coal is lower than that of high-rank coal. Coke B usually blends with a high proportion of low-rank coal. Thus, the proportion of aliphatic carbon bonds is high [27]. Nevertheless, coke A blends with a lower proportion of low-rank coal than coke B. As a result, coke B has a larger mass loss than coke An in the cohesive zone of the blast furnace.

**(2) Analysis of the microstructural changes in coke.**   The main structure of coking coal is a macromolecular polymer with an aromatic organic matter as its core. Its basic structural unit is a polyaromatic nucleus with side chain molecules. In coking, the side chains of aromatic nuclei continuously fall off and decompose. These nuclei shrink, condense, and form a microcrystalline structure, which is similar to a graphite crystal [28]. In the XRD spectra of coke, the (002) peak at 25˚ and (100) peak at 43˚ are detected. The (002) peak indicates the degree of the parallel and azimuthal orientations of aromatic carbon lamellae in microcrystals. A higher peak indicates better orientation of the aromatic carbon lamellae. The (100) peak represents the size of an aromatic carbon sheet. A higher peak corresponds to a higher condensation degree of aromatic nuclei [29,30].

In Fig 2, the (002) and (100) peaks increase after coke is heated to 1350˚C. The heights of the (002) peak of coke A before and after heating are 554 units and 581 units, respectively, so the peak height increases by 27 units. Furthermore, the heights of the (002) peak of coke B before and after heating are 376 units and 498 units, respectively, so the peak height increases by 122 units. The phenomenon of the (100) peak is similar to the previous result of the (002) peak. The heights of the (100) peak of coke A before and after heating are 90 units and 99 units, respectively, so the peak height increases by 9 units. The heights of the (100) peak of coke B before and after heating are 54 units and 69 units, respectively, so the peak height increases by 15 units. These results indicate that the degree of parallel orientation of aromatic lamellae and the degree of condensation of aromatic nuclei increase.

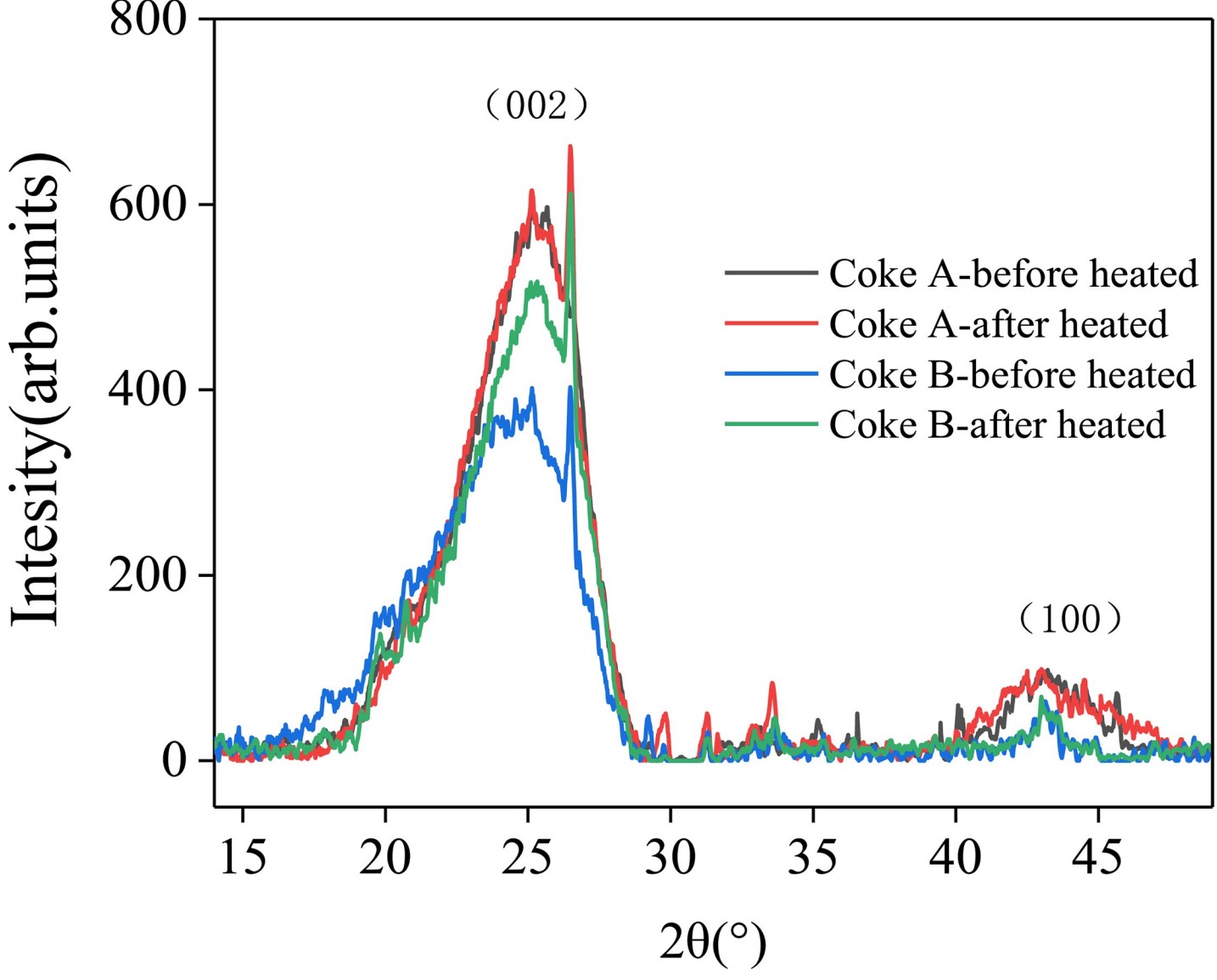

**Fig 2. XRD spectra of coke.**

Studies have shown that temperature is an important factor that affects the graphitization change in coke. In particular, this change slightly increases from 1000°C to 1200°C. Graphitization considerably changes when the temperature exceeds 1200°C [31,32]. Monaghan [33] and Gupta [34] studied the change in microcrystalline structure of coke after it was heated in a blast furnace. Carbon atoms tend to be orderly arranged after coke has been reheated in the blast furnace. The graphitization of coke is also deepened. Fi. 2 shows that the change in the (002) peak of coke B is more evident than that of coke A because more small molecules fall off from coke B than from coke A. This finding explains the more evident change in polymerization in aromatic nuclei than that in coke A. Therefore, the graphitization degree of coke increases when coke enters the cohesive zone, and the graphitization change in coke B is more evident than that in coke A.

Kejiang [35] demonstrated that the graphitization of coke caused the formation of coke fines on the coke surface. Their findings and the present results suggest that coke B, which is

clearly graphitized, produces more coke fines than coke A. The accumulation of coke fines is detrimental to the permeability of a coke bed.

### 3.2 Comparison of the gasification speed rate between coke A and coke B

The CRI indicates the percentage of total mass loss of coke after it reacts with $CO_2$ at 1100˚C. The temperature in the CRI test is notably different from the actual temperature in the blast furnace because the CRI cannot represent the actual gasification condition of coke in the blast furnace [14–16]. The temperature of the cohesive zone was simulated, and the gasification rate of coke was examined to study the gasification speed rate of coke in the blast furnace. The reaction curves of coke with different CRIs and $CO_2$ under the temperature condition (1100–1350˚C) in the cohesive zone are shown in Fig 3. Coke B loses mass more quickly than coke A. The reaction speed of coke with $CO_2$ is affected by temperature. The mass loss speed increases with temperature. The coke matrix determines the reaction speed rate. If coke contains a high-content isotropic matrix, the CRI of coke is high [36]. A higher CRI corresponds to a faster gasification rate. Coke quickly reacts with $CO_2$ in the cohesive zone of the blast furnace. The gasification rate of coke B is faster than that of coke A. This result indicates that the strength of

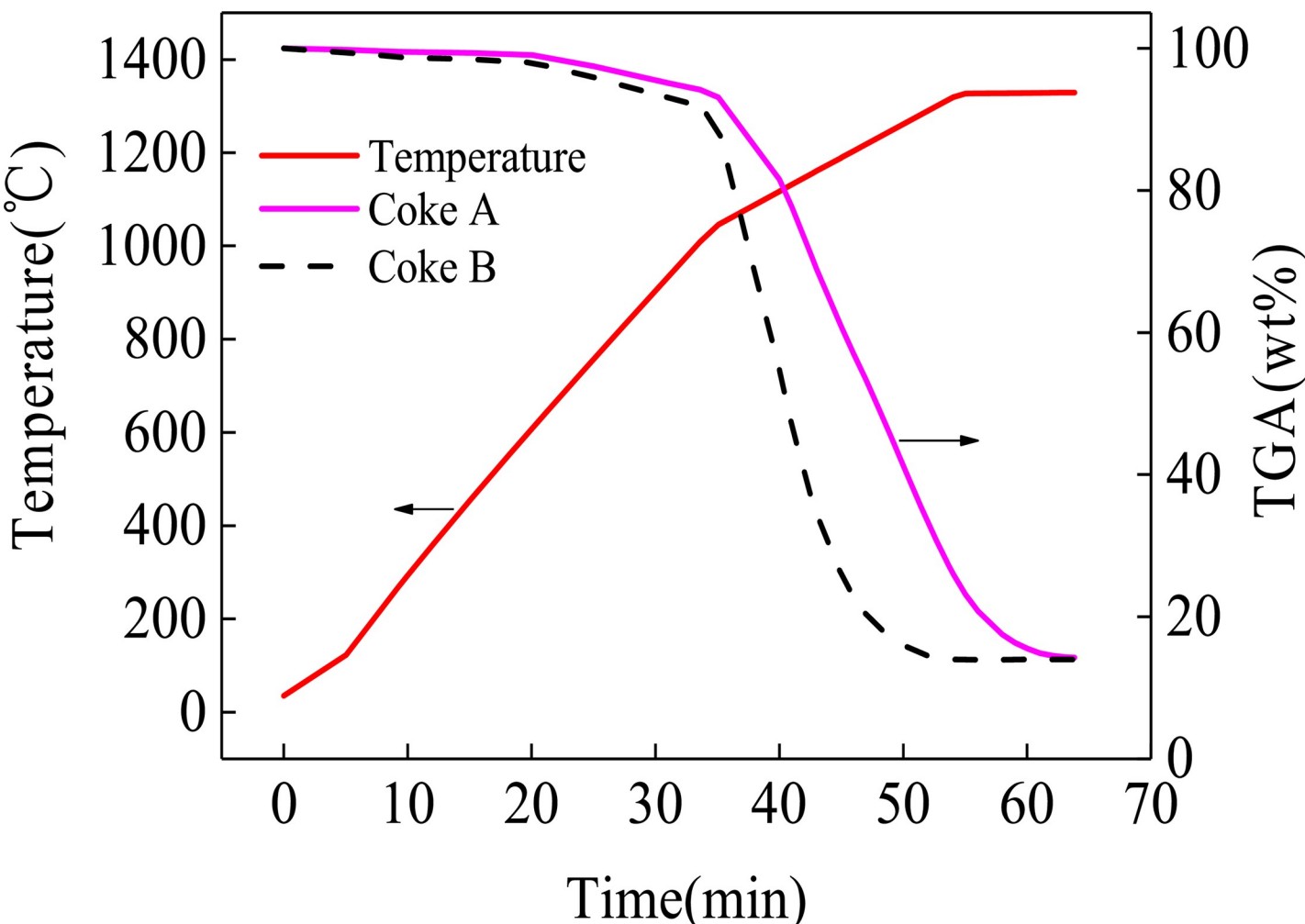

**Fig 3. Coke reaction with $CO_2$ at 1100–1350˚C.**

the former more quickly decreases than that of the latter because coke B loses more mass. Therefore, the pore wall of coke becomes thin, and its strength decreases. The gasification speed of coke at the cohesive zone temperature is consistent with that at the CRI test temperature. These results suggest that high-CRI coke more quickly degrades than low-CRI coke.

### 3.3 Reaction mechanism between coke and sinter

The reaction product images of coke A and coke B with the sinter are shown in Fig 4(A) and 4(B). The sinter completely melts and encloses coke A and B. Bubbles are also formed during the reaction.

Reaction processes can be analyzed by detecting gas signals [37]. In Figs 5 and 6, CO and $CO_2$ signals are simultaneously detected at 800˚C, which suggests that reduction occurs from 800˚C. In particular, the molten sinter directly reacts with coke to produce CO. CO quickly reacts with the sinter again and generates $CO_2$. The reaction time is extremely short, so CO and $CO_2$ simultaneously appear in Figs 5 and 6. The CO signal is stronger than the $CO_2$ signal because the degree of direct reduction between sinter and coke is more than that of indirect reaction between CO and sinter. CO does not have sufficient time to indirectly react with the sinter to produce $CO_2$, and CO quickly escapes from the reactant.

Another remarkable phenomenon is that both CO and $CO_2$ curves have two peaks at 1000˚C and 1200˚C. In comparison with the peaks in Fig 7, two peaks of the reaction speed rate curves are found at 1000˚C and 1200˚C. Iguchi [38] and Fang [39] found a hysteresis phenomenon in pellet reduction by hydrogen. Ubando [37] showed that the reduction in pellets had evident stages. The reduction mechanism can be summarized in three steps based on the temperature range. The first step follows a direct reduction via Eqs (3) and (4) at a temperature range of 800–1050˚C. Eqs (3) and (4) are the main reactions. Thermodynamically, $Fe_2O_3$ can

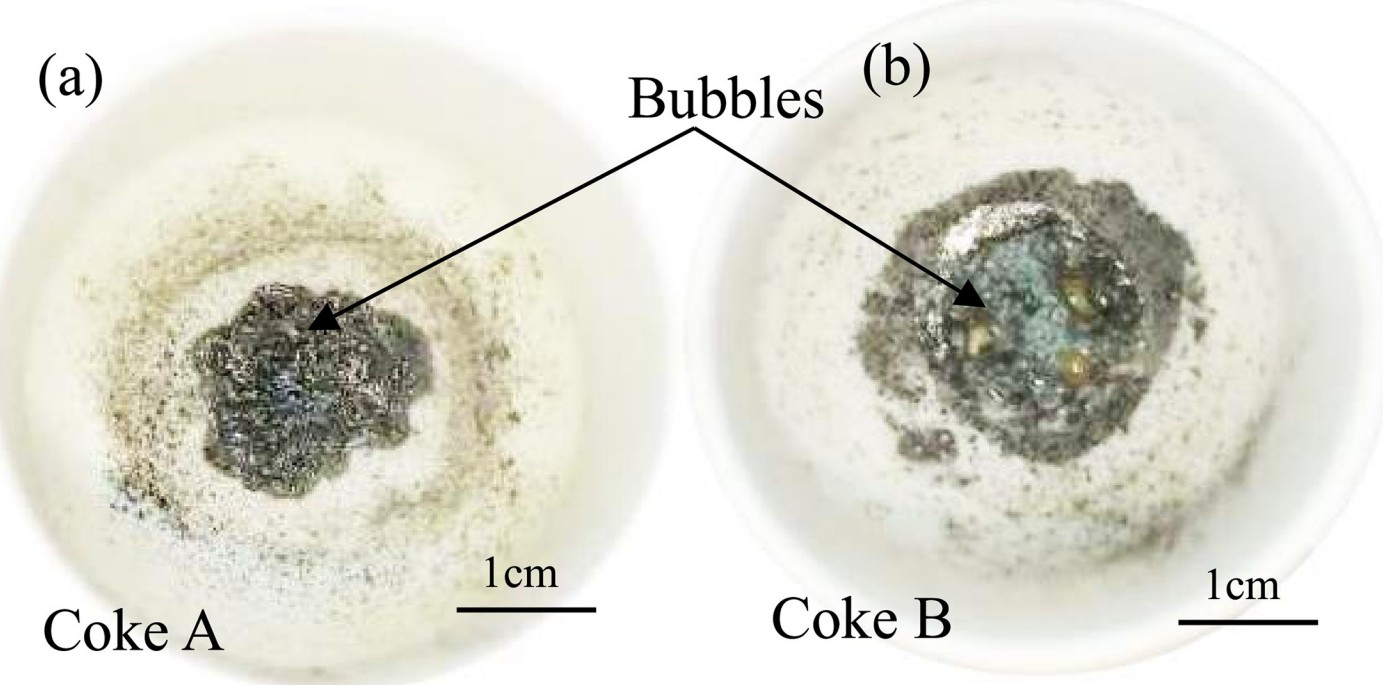

**Fig 4. Photograph of coke and sinter after the reaction.**

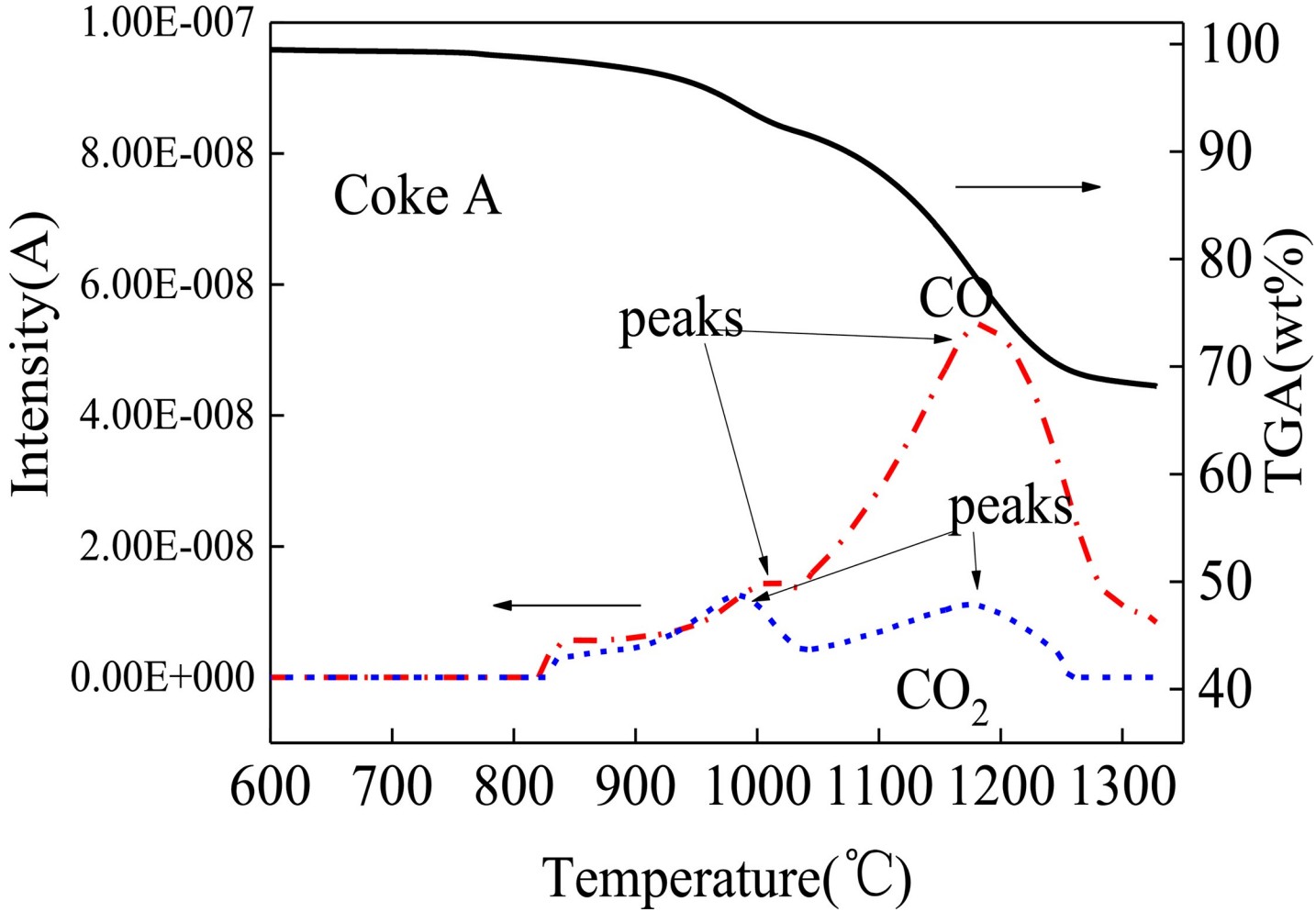

**Fig 5. Mass spectrogram of coke A reacting with sinter.**

be easily reduced by C and rapidly reduced to $Fe_3O_4$ to produce CO. This reaction reaches the peak at 1000°C. The CO peak implies that the reaction in Eq (4) also reaches the peak. Thus, $CO_2$ can be observed at the same temperature. C is consumed during direct reduction to generate CO. Side reactions such as Eqs (5)–(9) may also occur at low degrees. In the second step, Eqs (5) and (6) become the main reactions. Before the end of the $Fe_3O_4 \rightarrow FeO$ reaction, the third step in Eqs (7) and (8) has started. The two steps simultaneously proceed [40]. When the temperature reaches 1200°C, the reduction rate reaches the maximum speed point, as shown in Figs 5 and 6. The quantities of CO and $CO_2$ produced via oxidation reach the maximum point. Subsequently, the reactants are gradually depleted. When the temperature reaches 1300°C, the sinter is basically reduced to Fe. Fig 8(A) illustrates the reduced morphology of sinter. Fig 8(B) and Table 3 show point 1 chemical composition analysis, which can be used to infer the reduction degree of the sinter. The Fe and O contents in the reaction products are 83.92% and 7.71%, respectively. In addition, Al, Si, Ti and Zr are found in the products. These elements, which are typically in the form of their oxides in the coke ash and sinter, form a part of the slag from blast furnaces. The generation and reduction of slag in the cohesive zone are significantly affected by the burden containing iron. The sinter with a high content of alkalis is generally in the form of $Cao\text{-}SO_2\text{-}Al_2O_3\text{-}FeO\text{-}MgO$. The element analysis in Table 3 shows

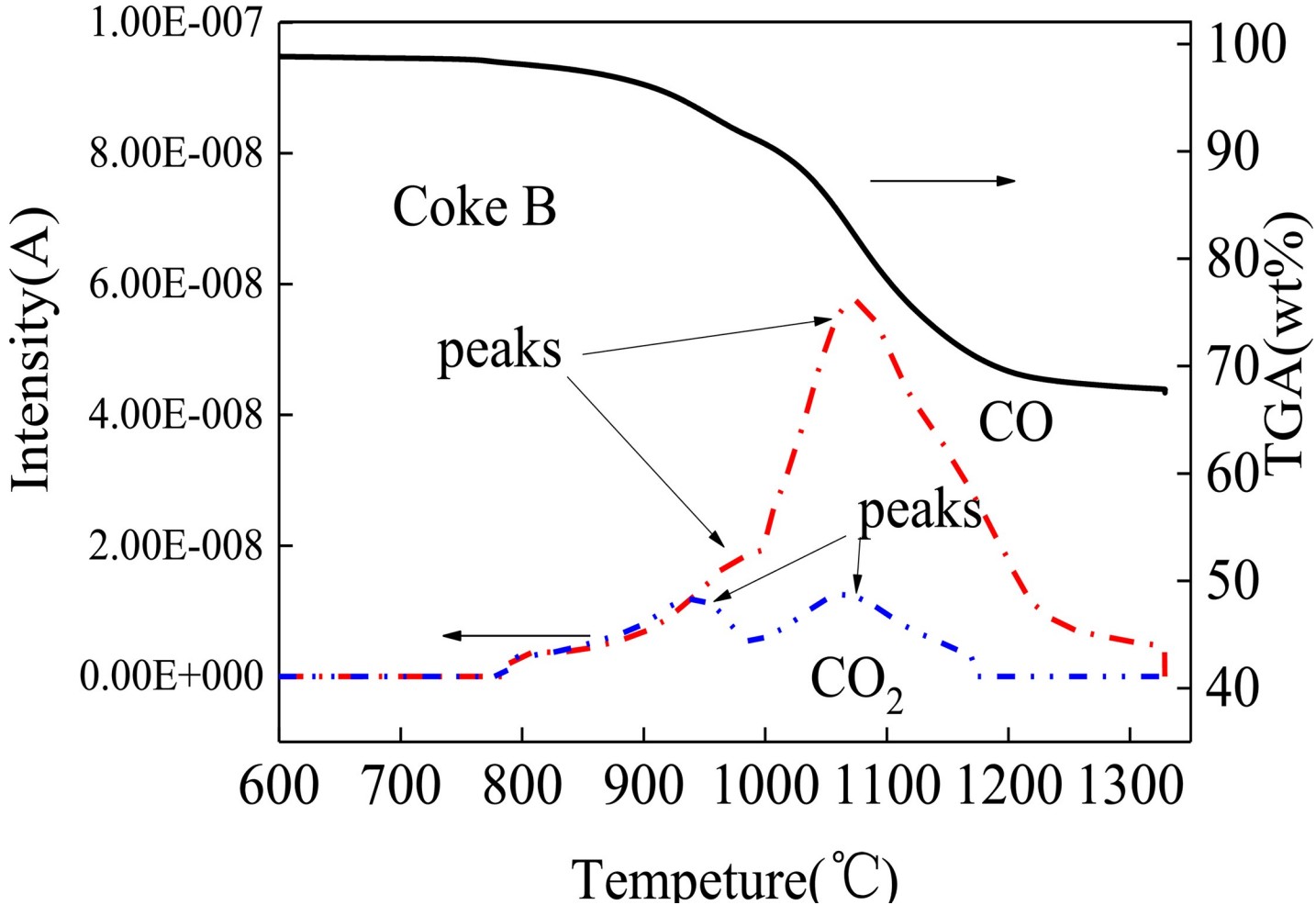

**Fig 6. Mass spectrogram of coke B reacting with sinter.**

that there are element differences between the slag in the present work and the primary slag from the blast furnace, which indicates that the ash and slag have not been fully integrated [41]. Thus, the same reaction steps occur in the cohesive zone of the blast furnace. However, depending on the conditions of different cohesive zones of a blast furnace, the compositions of reaction products differ.

$$3Fe_2O_3 + C \rightarrow 2Fe_3O_4 + CO \qquad (3)$$

$$3Fe_2O_3 + CO \rightarrow 2Fe_3O_4 + CO_2 \qquad (4)$$

$$Fe_3O_4 + C \rightarrow 3FeO + CO \qquad (5)$$

$$Fe_3O_4 + CO \rightarrow 3FeO + CO_2 \qquad (6)$$

$$FeO + C \rightarrow Fe + CO \qquad (7)$$

$$FeO + CO \rightarrow Fe + CO_2 \qquad (8)$$

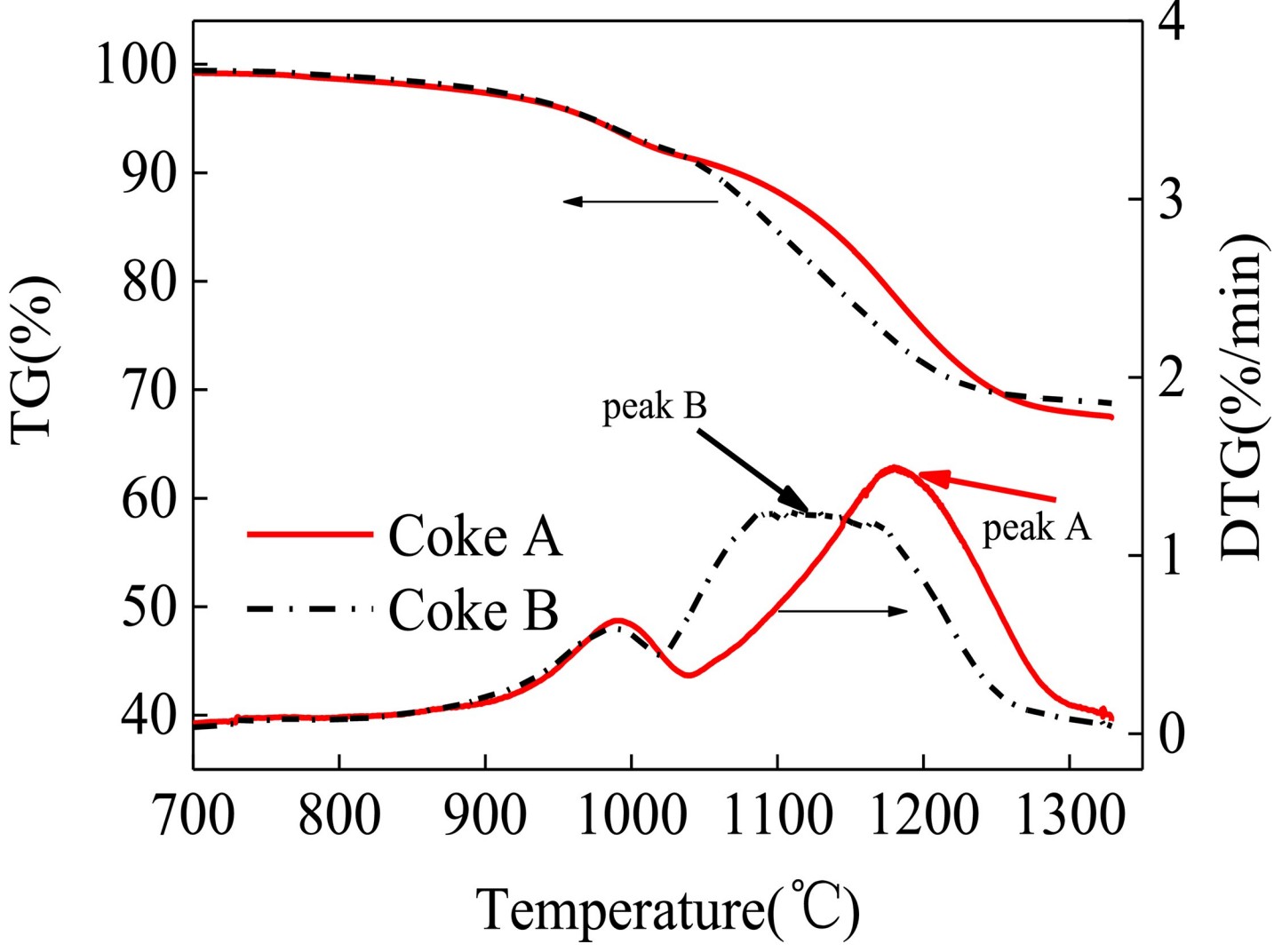

**Fig 7. DTG curves of coke reacting with sinter.**

$$CO_2 + C \rightarrow 2CO \qquad (9)$$

### 3.4 Differential thermal analysis between coke and sinter

As shown in Figs 9 and 10, there are endothermic peaks near 1000–1200°C in the reaction process between coke and sinter. The endothermic peak of coke A is significantly higher than that of coke B. Coke A reacts with sinter to absorb more heat than coke B because coke A has a higher molecular bond energy than coke B, which causes coke to absorb more energy when it reacts with sinter. The molecule is separated from the carbon molecular skeleton.

### 3.5 Comparative analysis of reaction speed rates between coke and molten sinter

When the temperature is approximately 1000°C, coke A and coke B reach the curve peaks of their first reaction speed rate. The reaction speed rates of coke A and B are similar, as shown in Fig 7. When the temperature exceeds 1000°C, coke B and coke A reach their maximum speed

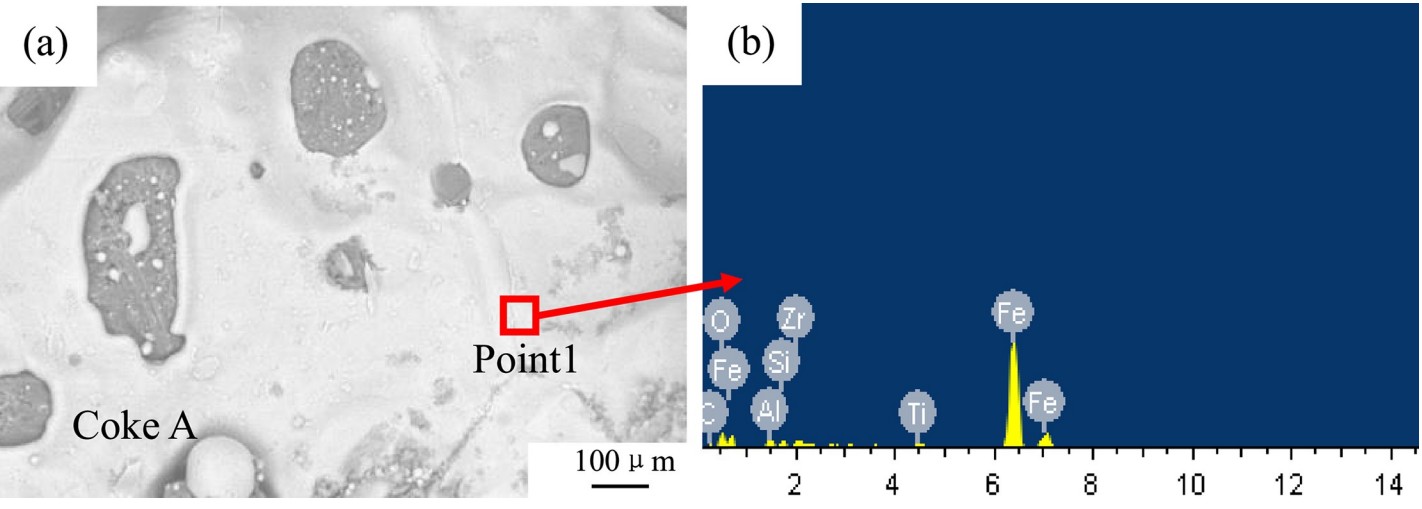

**Fig 8. EDS images of the morphological characteristics of the sample.**

values near 1100˚C and 1200˚C, respectively. Then, coke A and coke B are gradually consumed, and the reaction is finished. This result shows that the speed rate curve of coke B reaches the maximum before that of coke A, which indicates that the carbon molecule of coke B is more active and easier to dissociate from the carbon skeleton than that of coke A. Wang [42] believed that coke B was beneficial to sinter reduction. The chemical bond energy between low-CRI molecule fractions is high, which requires high temperature and energy to react with sinter. Thus, coke B can be easily consumed by the sinter. The consumption of coke leads to particle size reduction, which is not conducive to the gas and liquid permeability of the cohesive zone. The rapid decrease in particle size of coke B is not conducive to the operation of the cohesive zone.

Coke A has a higher maximum reaction speed rate than coke B because of two reasons. First, coke B reacts earlier with sinter than coke A, so there is less residual reactant in coke B than in coke A (Fig 8). When coke A reaches the maximum peak temperature, the base of the residual reactant reflects the consumption rate. Second, the consumption speed rate of coke B is more uniform, and its reaction speed rate peak is gentler with a high reaction speed rate in the temperature range of 1100–1200˚C. However, the peak reaction speed rate of coke A is sharper, which indicates that the carbon dissociation speed is not uniform. When temperature reaches the dissociation temperature of carbon molecules, they simultaneously dissociate from the skeleton. Thus, coke A has a higher maximum reaction speed rate than coke B.

**Table 3. Elemental analysis.**

| Elements | Weight percentage/% | Atomic percentage/% |
|---|---|---|
| C | 0.67 | 2.51 |
| O | 7.71 | 21.73 |
| Al | 2.08 | 3.48 |
| Si | 1.16 | 1.87 |
| Ti | 1.07 | 1.01 |
| Fe | 83.92 | 67.74 |
| Zr | 3.39 | 1.67 |

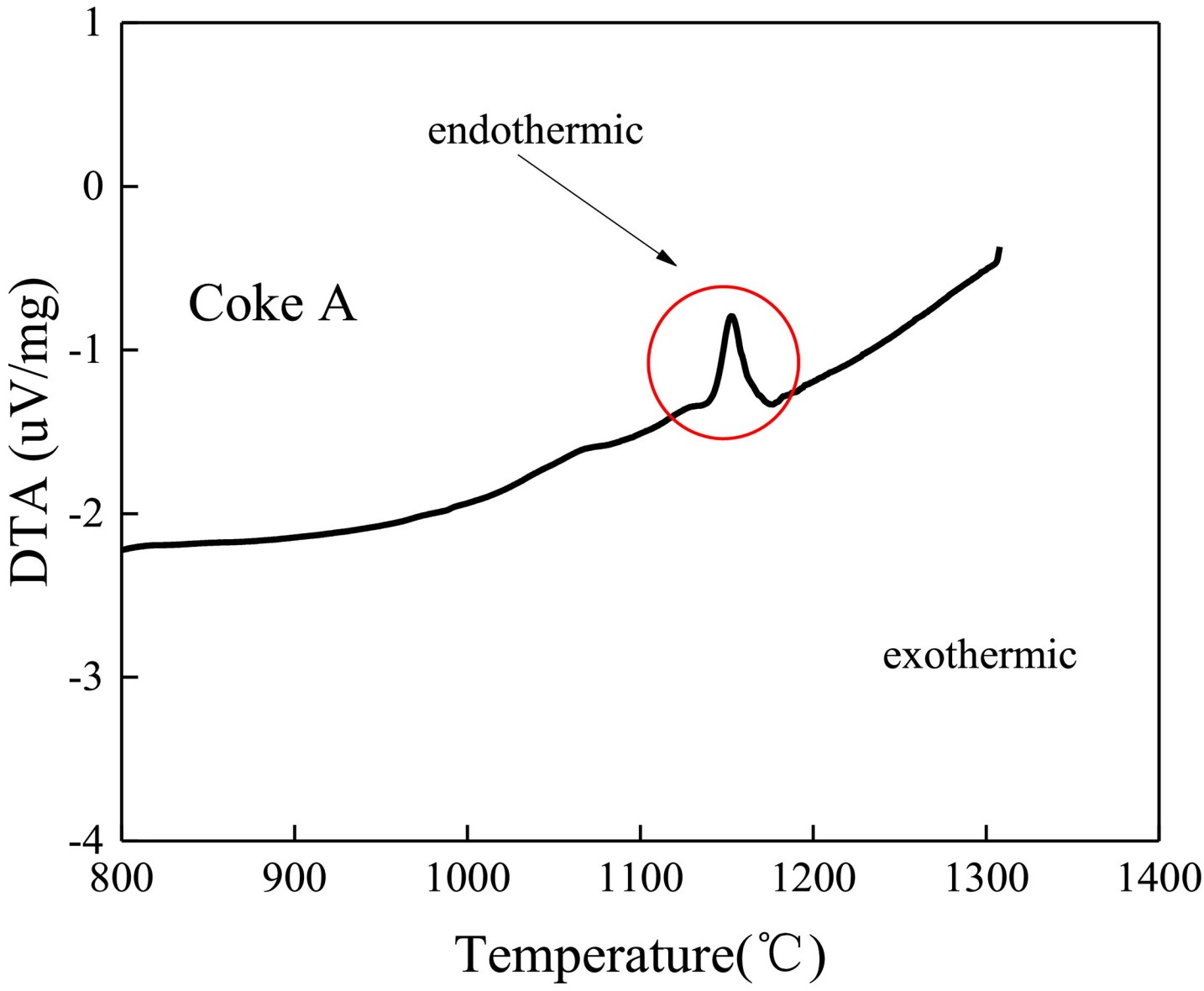

**Fig 9. Thermal spectrogram of coke A reacting with sinter.**

## 3.6 Characteristics of the reaction interface between coke and molten sinter

Fig 11(A)–11(D) illustrate the interfaces between coke and sinter. As shown in Fig 11(A) and 11(B), both coke samples are encapsulated. Table 3 indicates that the sinter is reduced to iron. In Fig 11(C) and 11(D), many pits are formed on the surfaces of coke A and B during the reaction. This result reveals that the size of coke decreases after it is consumed by the sinter in the cohesive zone of the blast furnace. Iron contains C (Table 3), which suggests that sinter reduction and carburization are almost simultaneous. The reaction surfaces of coke A and coke B with sinter are nearly identical. Therefore, the consumption of the coke surface by sinter is identical.

Chapman [43] studied the dissolution of carbon into molten iron. Sahajwalla et al. [44,45] revealed the dissolution of graphite carbon in molten iron at an atomic level through simulation. The contact between solid graphite carbon and molten iron causes the dissociation of

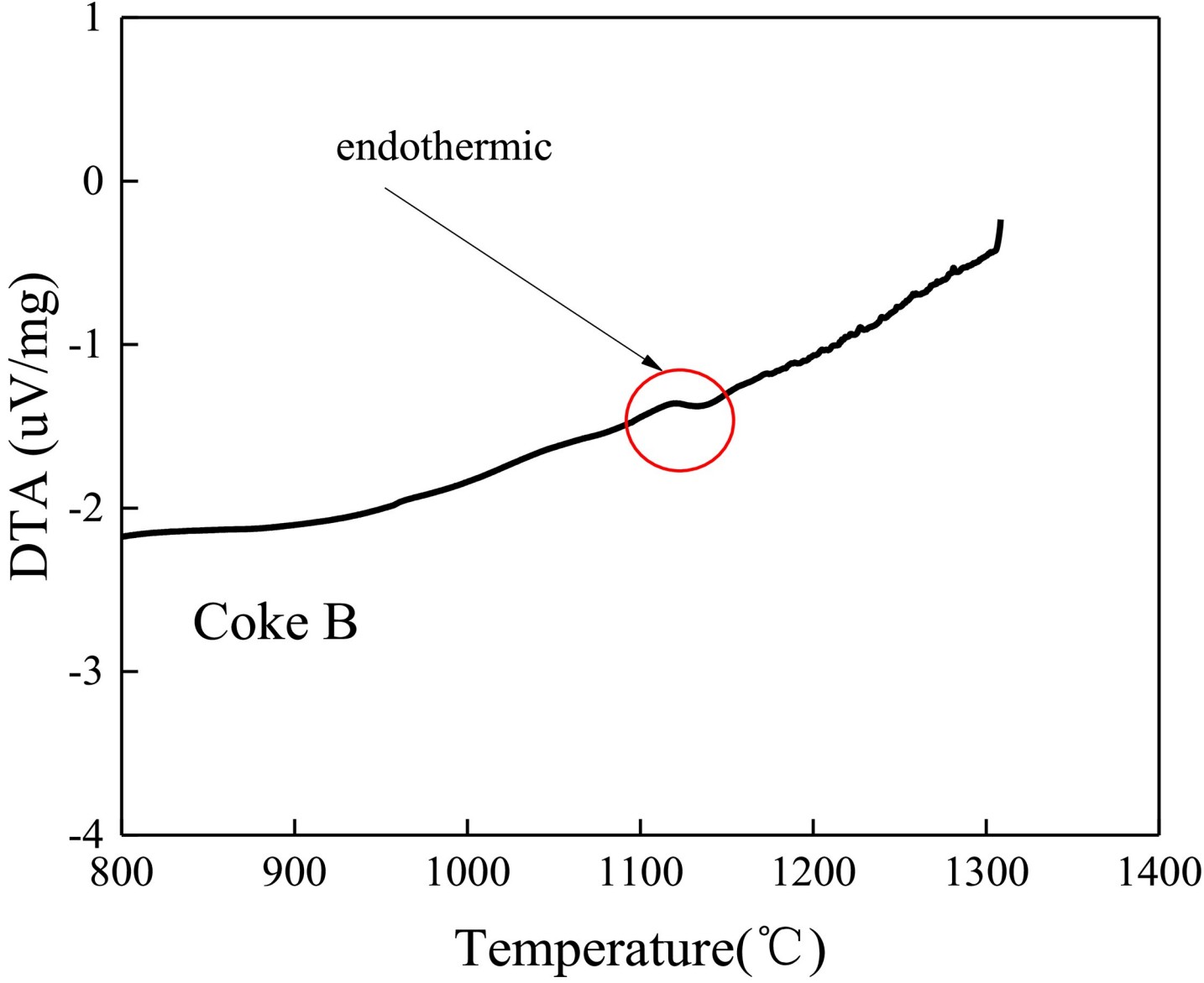

**Fig 10. Thermal spectrogram of coke B reacting with sinter.**

carbon atoms from a graphite microcrystalline structure. Coke starts to carburize into molten iron in the cohesive zone of the blast furnace. The dissociation speed rate of carbon atoms is faster than the transfer rate of carbon atoms in molten iron. Therefore, carbon pyrolysis is considered a first-order dynamic process, and its limiting link is the process of carbon mass transfer in molten iron [46]. Fig 7 shows that the carbon molecule of coke B is more active and easier to dissociate from the carbon skeleton than that of coke A, so the carbon atoms of coke B easily transfer to molten iron.

## 4 Conclusions

In this study, coke A and coke B were selected to study the reaction behavior of coke in the cohesive zone of a blast furnace through thermogravimetric analysis. Their microstructural changes

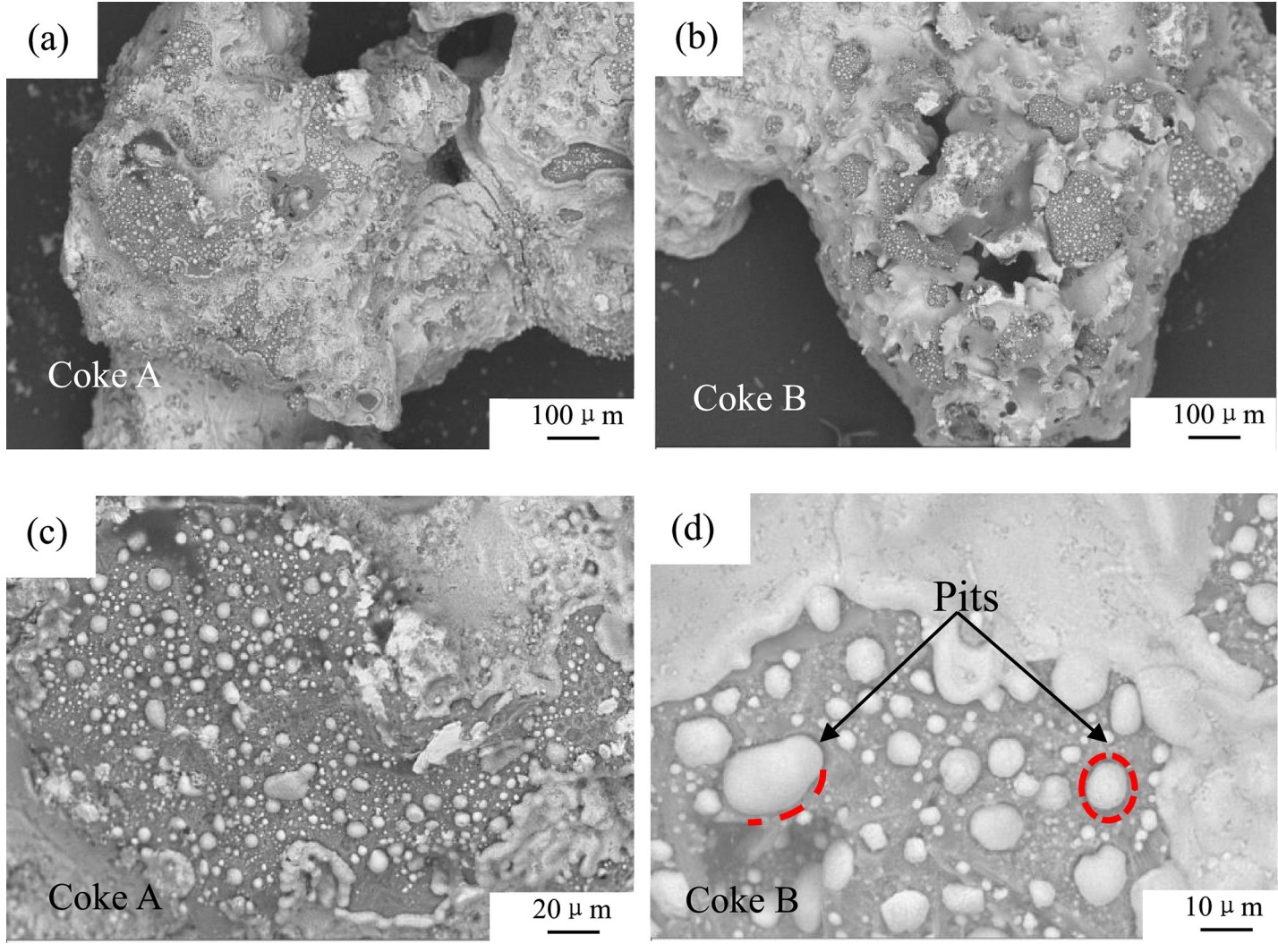

**Fig 11. Surface morphology of coke.**

and reaction mechanism were analyzed. Under the cohesive zone temperature condition in the blast furnace, coke A and coke B underwent further pyrolysis and polycondensation. Direct and indirect reduction reactions occurred between coke and sinter in the cohesive zone. The graphitization, gasification, and consumption rates of coke B are higher than those of coke A, which indicates that coke B has faster degradation speed than coke A. The difference in CRI lies in the difference in molecular bond energy. According to the graphitization, gasification, and consumption rate research, the CRI of coke plays a vital role in the deterioration of coke in the cohesive zone of a blast furnace. An in-depth understanding about the reactions of coke with different CRIs in the high-temperature zone of a blast furnace requires further research. S1(A) and S1(B) Fig in S1 File show that slag and iron enter the inner holes of coke through open pores.

## Supporting information

**S1 File.**
(ZIP)

## Author Contributions

**Conceptualization:** Hua W. Ren.

**Data curation:** Yong S. Tian.

**Formal analysis:** Hua W. Ren.

**Investigation:** Jun L. Zhou.

**Methodology:** Guang H. Wang.

**Writing – original draft:** Qing Q. Lv.

**Writing – review & editing:** Guang H. Wang.

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
