## [Decision Letter · Decision Letter 0]

14 Oct 2020

PONE-D-20-19167

A Comprehensive Investigation on Reaction Behavior Features of Coke with Different CRI in the Simulated Cohesive Zone of a Blast Furnace

PLOS ONE

Dear Dr. Lv,

Thank you for submitting your manuscript to PLOS ONE. After careful consideration, we feel that it has merit but does not fully meet PLOS ONE’s publication criteria as it currently stands. Therefore, we invite you to submit a revised version of the manuscript that addresses the points raised during the review process.

I have received reports from both reviewers. Now I can have feedback on your manuscript and on this basis, I can assure you that your paper can be accepted for publication in PLOS ONE journal, once the attached comments, concerns and feedback are properly addressed in the next version of the paper. Please ensure the revised version to be free of typos and grammatical errors. Also, attach a rebuttal letter consisting of your responses against each comment.

Looking forward to receiving the revision.

We look forward to receiving your revised manuscript.

Kind regards,

Dr Mohsen Sarafraz

Academic Editor

PLOS ONE

Journal Requirements:

3. Please provide further information on the two types of coke used in the study, such as characteristics, brand/vendor information and how the coke was obtained.

4. Thank you for stating the following in the Financial Disclosure section:

'Guang-hui Wang received for this work.

This work was financially Supported by National Key R&D Program of China (2017YFB0304300&2017YFB0304303)

No sponsors of funder play any role in the study design, data dollection and analysis, desision to publish, or preparation of the manuscript.'

We note that one or more of the authors are employed by commercial companies:

Key Laboratory of Hubei Province for Coal Conversion and New Carbon Materials, Iron and Steel Research Institute of Shasteel, Zhangjiagang and Xingtai Risun Science and Technology Co., Ltd.

Please amend your Competing interests statement to declare author commerical affiliations.

5. Please include captions for your Supporting Information files at the end of your manuscript, and update any in-text citations to match accordingly. Please see our Supporting Information guidelines for more information: http://journals.plos.org/plosone/s/supporting-information

Additional Editor Comments:

Dear authors

I have received reports from both reviewers. Now I can have feedback on your manuscript and on this basis, I can assure you that your paper can be accepted for publication in PLOS ONE journal, once the attached comments, concerns and feedback are properly addressed in the next version of the paper. Please ensure the revised version to be free of typos and grammatical errors. Also, attach a rebuttal letter consisting of your responses against each comment.

Looking forward to receiving the revision.

Reviewers' comments:

Reviewer's Responses to Questions

**Comments to the Author**

1. Is the manuscript technically sound, and do the data support the conclusions?

Reviewer #1: Yes

Reviewer #2: Yes

2. Has the statistical analysis been performed appropriately and rigorously? 

Reviewer #1: Yes

Reviewer #2: N/A

3. Have the authors made all data underlying the findings in their manuscript fully available?

Reviewer #1: Yes

Reviewer #2: Yes

4. Is the manuscript presented in an intelligible fashion and written in standard English?

Reviewer #1: Yes

Reviewer #2: Yes

5. Review Comments to the Author

Reviewer #1: The authors present a study on the microstructural changes, gasification characteristics and reaction mechanism of coke with different CRI in a thermogravimetric analyzer. The fundamental characteristics were studied and tested by using different techniques, such as TGA, DTG, DTA, XRD, optical microscope, SEM and EDS etc. After reading this paper some questions arise that need be clarified by the authors.

1) In Table 1, the words CRI and CSR were explained in the manuscript, but the meaning of the words M40, M10, Vad, St,d and Ad was not appeared. Are these words familiar to professionals?

2) In Table 2,

Table 2 Chemical composition of sinter (wt. %)

Elements Fe FeO SiO2 Al2O3 CaO MgO P S

Sinter 57.62 9.62 4.55 1.85 10.28 1.71 0.059 0.018

What is the meaning of letter T before Fe?

3) In page5 paragraph 2, “In Fig. 2, (002) and (100) peaks increase when coke is heated to 1350 ℃.” Should be “In Fig. 2, (002) and (100) peaks increase after coke is heated to 1350 ℃.”.

4) In Fig. 4, the scale should be added, even in the macrography.

5) References 1-23 are repeated.

6) The format of references should be unified, such as the references 4, 13, 16, 17, 23, 28, 36 and 41 should be use the full name of the journal, because the full name is used in other references; there is one more 2007 in Ref. 7; the name of the last author in Ref. 26.

Reviewer #2: Overall the manuscript “A Comprehensive Investigation on Reaction Behavior Features of Coke with Different CRI in the Simulated Cohesive Zone of a Blast Furnace” is a well described study investigating the reaction behavior of coke in blast furnace. I have few questions and comments which needs to be answered.

1. Equation 1 and two looks very shrink. It can be modified

2. Why elemental analysis was carried out in this study and what were the significance of analyzing only those elements which are presented in the table 3.

3. Conclusion should be summarized in single paragraph rather than in points and should not be very long. It must reflect the findings and future perspectives of the study.

4. Figure 2 is not very clear. One cannot differentiate among different peaks.

5. Some of the latest references should be added like 2015 onward.

6. Author should add some figures to the supplementary data as with so many figures in the main text will make manuscript very crowded.

6. PLOS authors have the option to publish the peer review history of their article (what does this mean?). If published, this will include your full peer review and any attached files.

Reviewer #1: **Yes: **Xueqin Kang

Reviewer #2: No

---

## [Author Response · Author response to Decision Letter 0]

21 Dec 2020

Dear two reviewers, 

It is a great honor to receive the comments on this manuscript. Thank you for your patience. All the comments are very professional. I learned a lot from it. I have revised the manuscript to meet the requirements and the format of PLOS ONE. And I have chosen AJE for language editing of my manuscript. I hope that the resubmitted manuscript can meet the requirements of PLOS ONE.

If the manuscript still needs to be improved, please give feedback in time, and I will make corresponding modifications.

Thank you very much.

---

## [Editor Report · Decision Letter 1]

23 Dec 2020

A Comprehensive Investigation of the Reaction Behaviorial Features of Coke with Different CRIs in the Simulated Cohesive Zone of a Blast Furnace

PONE-D-20-19167R1

Dear Dr. Lv,

We’re pleased to inform you that your manuscript has been judged scientifically suitable for publication and will be formally accepted for publication once it meets all outstanding technical requirements.

Kind regards,

Dr Mohsen Sarafraz

Academic Editor

PLOS ONE
---

## [Editor Report · Acceptance letter]

28 Dec 2020

PONE-D-20-19167R1 

A Comprehensive Investigation of the Reaction Behaviorial Features of Coke with Different CRIs in the Simulated Cohesive Zone of a Blast Furnace 

Dear Dr. Lv:

I'm pleased to inform you that your manuscript has been deemed suitable for publication in PLOS ONE. Congratulations! Your manuscript is now with our production department. 

Kind regards, 

on behalf of

Dr. Dr Mohsen Sarafraz 

Academic Editor

PLOS ONE